# Spatial Analysis of NQO1 in Non-Small Cell Lung Cancer Shows Its Expression Is Independent of NRF1 and NRF2 in the Tumor Microenvironment

**DOI:** 10.3390/biom12111652

**Published:** 2022-11-08

**Authors:** Boback Kaghazchi, In Hwa Um, Mustafa Elshani, Oliver J. Read, David J. Harrison

**Affiliations:** 1School of Medicine, University of St Andrews, St Andrews KY16 9TF, UK; 2NuCana plc, 3 Lochside Way, Edinburgh EH12 9DT, UK

**Keywords:** reactive oxygen species, non-small cell lung cancer, tumor microenvironment

## Abstract

Nuclear factor erythroid 2-related factor 1 (NFE2L1, NRF1) and nuclear factor erythroid 2-related factor 2 (NFE2L2, NRF2) are distinct oxidative stress response transcription factors, both of which have been shown to perform cytoprotective functions, modulating cell stress response and homeostasis. NAD(P)H:quinone oxidoreductase (NQO1) is a mutual downstream antioxidant gene target that catalyzes the two-electron reduction of an array of substrates, protecting against reactive oxygen species (ROS) generation. NQO1 is upregulated in non-small cell lung cancer (NSCLC) and is proposed as a predictive biomarker and therapeutic target. Antioxidant protein expression of immune cells within the NSCLC tumor microenvironment (TME) remains undetermined and may affect immune cell effector functions and survival outcomes. Multiplex immunofluorescence was performed to examine the co-localization of NQO1, NRF1 and NRF2 within the tumor and TME of 162 chemotherapy-naïve, early-stage NSCLC patients treated by primary surgical resection. This study demonstrates that NQO1 protein expression is high in normal, tumor-adjacent tissue and that NQO1 expression varies depending on the cell type. Inter and intra-patient heterogenous NQO1 expression was observed in lung cancer. Co-expression analysis showed NQO1 is independent of NRF1 and NRF2 in tumors. Density-based co-expression analysis demonstrated NRF1 and NRF2 double-positive expression in cancer cells is associated with improved overall survival.

## 1. Introduction

Lung cancer is the leading cause of tumor-related mortality worldwide, accounting for approximately 20% of all cancer-related deaths [1]. It can be divided into two broad types, small cell lung cancer (~15%) and non-small cell lung cancer (NSCLC, ~85%). NSCLC is classified within three broad subtypes: adenocarcinoma, squamous carcinoma and large-cell carcinoma, according to its proposed cell type of origin within the lung [2]. NSCLC has a poor five-year survival, presenting with a SEER (surveillance, epidemiology, and end result) stage combined five-year relative survival rate of 26%, primarily relating to late-stage diagnosis [3,4]. Lung adenocarcinoma has been reported to account for over 38.5% of total lung cancer cases [5]. Lung adenocarcinoma constitutes a group of molecularly and histologically heterogeneous diseases within the same histological subtype, displaying significant differences between key cancer hallmarks such as immune checkpoint modulation, DNA repair mechanisms and redox status [6,7]. The tumor microenvironment (TME), comprised of various cell types, including disseminated aggressive and proliferating tumor cells from the mass, tumor stroma, blood vessels and tumor-infiltrating immune cells (TILs), contribute to cancer progression [8]. Advanced NSCLC patients have been demonstrated to display heterogeneity in tumor cells’ cellular composition, intracellular signaling networks and TME cells’ interaction network, such as tumor-associated neutrophils and macrophages [9]. These patterns of heterogeneity within the tumor and the tumor microenvironment have recently re-conceptualized the hallmarks of cancer, particularly the relationship of redox with cancer [10].

An altered redox state and dysfunctional antioxidant capacity are often observed in lung cancer owing to concomitant cellular processes, such as metabolic changes observed during pathogenesis [8,11]. Oxidative stress results from an imbalance of reactive oxygen species (ROS) and a cell’s antioxidant capacity. Management of ROS is required for many physiological processes; however, dysregulation of ROS has been implicated in lung cancer pathogenesis, causing damage to proteins, nucleic acids and other molecules via oxidation, affecting their homeostatic function [12]. Within the TME, excessive ROS can contribute to immunosuppressive mechanisms utilized by suppressive cells against immune effectors such as CD8^+^ cytotoxic T-cells, which facilitate the anticancer response and are the cornerstone of modern successful cancer immunotherapies [13]. ROS molecules and peroxynitrite derived from myeloid-derived suppressor cells (MDSCs) have been demonstrated to result in nitration of T-cell receptors and CD8 molecules, causing conformational changes in these molecules, thereby decreasing CD8^+^ cytotoxic T-cell ability to bind phosphorylated major histocompatibility complex (MHC) and allow for antigen-specific tolerance of peripheral CD8^+^ T cells [14].

The nuclear factor erythroid 2-related factor 1 (NFE2L1) and nuclear factor erythroid 2-related factor 2 (NFE2L2), also referred to as NRF1 and NRF2 respectively, are two distinct oxidative stress response transcription factors belonging to the cap’n’collar (CNC)/basic-region leucine zipper (bZIP) protein family which affect cellular stress response survival, homeostasis and development [15,16]. NRF1- and NRF2-signaling pathways protect against oxidative damage and other stress instituted by increased ROS levels by upregulating antioxidant genes through antioxidant response element (ARE) binding, including glutamate-cysteine ligase catalytic (GCLC), heme oxygenase 1 (HO-1) and NAD(P)H: quinone oxidoreductase 1 (NQO1). NRF3 (NFE2L3) is a less-described CNC-basic leucine zipper transcription factor with functions not yet fully characterized due to the lack of robust commercial antibodies to progress research further. Known functions of this protein include the induction of 20S proteasome assembly genes and modulation of cell cycle progression via cyclin-dependent kinase 1, thereby driving colon cancer cell proliferation [17]. NRF3 is upregulated in a variety of cancer tissues when compared to normal tissue, including head and neck squamous cell carcinoma and colon cancer [17,18].

Current literature suggests that NRF1 regulates basal forms of oxidative stress and regulates proteasome subunit expression, while the ‘master regulator of the antioxidant response’ NRF2 is primarily responsive towards inducible oxidative stress. In the CNC/bZIP protein family, NRF1 and NRF2 are the best-described transcription factors, both expressed ubiquitously in vertebrate tissues and demonstrated overlapping and competitive functions in cell and mouse models [19]. One mutual downstream NRF1 and NRF2 antioxidant gene target in mice is nqo1 [20]. NQO1 is a detoxification enzyme that catalyzes the two-electron reduction of an array of substrates, protecting against the generation of ROS [21]. It has been reported that NQO1 has high expression in adipocytes, epithelium and solid tumors. Recent studies report that NQO1 is upregulated in NSCLC in contrast to adjacent non-tumor tissue and is proposed to be a poor prognostic biomarker in stage I-II lung cancer and a potential therapeutic target for NSCLC patients [22]. Conversely, the NQO1 expression in stage III–IV patient samples from the same cohort did not correlate with the patients’ overall survival rate. However, Siegel and colleagues previously described high levels of NQO1 expression in the respiratory epithelium, including bronchus and capillaries, hypothesizing that NQO1 could exert a protective effect upon the epithelium [23].

NRF2 has been reported to protect cells and nucleic acids from oxidative stress during the early stages of tumorigenesis; however, in later stages, NRF2 contributes to chemoresistance, promoting carcinogenesis and metastasis [24,25]. Additionally, a high frequency of somatic loss-of-function mutation of Kelch Like ECH Associated Protein 1 (KEAP1), a negative regulator of NRF2, has been reported in NSCLC [26]. Importantly, NRF2 has previously been shown to inhibit the immunosuppressive actions of MDSCs in mouse models by eliminating ROS and allowing for tumor immunity by protecting regulatory T-cells, which are vulnerable to oxidative stress [27,28]. In a mixed NSCLC subtype cohort, Tong and colleagues reported that NQO1 and NRF2 are not individual prognostic factors [29]. However, stage III and IV dual-negative NQO1 and NRF2 patients displayed better overall survival, with no significant differences seen in earlier stages (I and II).

Increased proteasomal activity has been reported in cancer cells, potentially due to the increased proteotoxic stress and protein homeostasis burden linked to cancer transformation [30]. NRF1 has been demonstrated to regulate proteasome subunit transcription. This is shown by proteasomal inhibition, in which a “bounceback” response has been evidenced via NRF1 nuclear translocation and proteasomal target gene expression [31]. 

NRF1 is a potential therapeutic target, displaying differential expression in prostate cancer tumor tissue versus adjacent non-tumor tissue [32]. High NRF1 expression is a good prognostic factor in renal and endometrial cancer; however, NRF1 remains to be investigated in a NSCLC cohort (Human Protein Atlas). NRF1- and NRF2-mediated nqo1 transcriptional upregulation dictates that co-localization of these three key antioxidant proteins requires further examination within cancer cells and the tumor microenvironment to determine whether NRF1, NRF2 and NQO1 co-expression is important in NSCLC pathogenesis and to investigate the potential patient outcome value of such co-expression [20]. Previous NSCLC NQO1 studies utilized mixed subtype cohorts and performed IHC and immunofluorescence with single antibodies targets, demonstrating differences in antioxidant function and prognostic outcome, which is dependent on the cancer stage [22,29]. Therefore, we aimed to address the findings of NQO1 expression across the TME and normal tissue, co-registered with NRF1 and NRF2 in lung adenocarcinoma to expand NRF1 and NRF2 research in NSCLC.

## 2. Materials and Methods

### 2.1. Patients

Samples for this study were collected from patients with early-stage non-small cell lung cancer treated by primary surgical resection at the Royal Infirmary of Edinburgh with NHS Research Scotland Lothian Bioresource approval (Lothian 10/S1402/33). This cohort has previously been described [33]. Inclusion criteria were: (1) patients with NSCLC of adenocarcinoma histology of stage I–IIa disease, who, in accordance with the Seventh Edition International Association for the Study of Lung Cancer (IASLC) guidelines, (2) underwent complete resection of the primary treatment, having received no radiotherapy or chemotherapy before surgery and no adjuvant treatment with radiation or chemotherapy within 12 weeks of surgery and during the one-month follow-up period. Patients previously diagnosed with lung cancer or synchronous lung cancers were excluded. The primary outcome, a measure of death from lung cancer, was determined by clinical records and the secondary outcome measure, overall survival, was also reviewed. A total of 162 samples were utilized for this study based on tissue block availability and sample quantity remaining. All samples were deidentified to researchers to keep laboratory analysis and assays blinded to clinical and pathological data, with survival at five years taken as the cut-off point.

### 2.2. Multiplexed Immunofluorescence (mIF) and Image Acquisition

Formalin-fixed paraffin-embedded (FFPE) blocks were sectioned at 2.5 μm and dried at 65 °C. Tissue sections were deparaffinized in xylene and rehydrated using an ethanol concentration gradient -100%, 100%, 80%, and 50%-and running tap water. The epitopes in cells were unmasked using pH 6 sodium citrate buffer for 5 min in a pressure cooker. Sections were washed in Tris-buffered saline with 0.1% Tween^®^ 20 Detergent (TBST) buffer, and endogenous peroxidase activity and non-specific background stain were blocked by 3% hydrogen peroxide (Sigma, #H1009, Kawasaki, Japan) and serum-free protein blocking buffer (Agilent, #X090930-2, Santa Clara, CA, USA), respectively. The first primary antibody, NRF1 (Human Protein Atlas, #HPA065424, 1:1500), was incubated for 1 h at room temperature, followed by the secondary antibody, anti-rabbit HRP (Agilent, #K400311-2), for 1 h. NRF1 was visualized by TSA Cyanine 3 (Akoya Bioscience, #NEL744001KT, 1:50, Marlborough, MA, USA). Then, the sections were treated with pH6 sodium citrate buffer having boiled for 17 min in a microwave to remove any redundant antibodies. 

The second and third primary antibodies, (NQO1 (Human protein atlas, #HPA007308, 1:800) and NRF2 (ProteinTech, #16396-1-AP, 1:2000, Rosemont, IL, USA), visualized with by TSA fluorescein (Akoya Bioscience, #NEL741001KT, 1:50) and TSA Cyanine 5 (Akoya Bioscience, #NEL745001KT, 1:50), respectively. Lastly, to visualize epithelial cells, pan-cytokeratin AE1/AE3 (Agilent, #M351501-1, 1:100) was incubated and visualized by avidin-biotin reaction using an anti-mouse biotinylated secondary antibody (ThermoFisher, #31800, 1:25, Greenville, NC, USA), followed by streptavidin-conjugated Alexa Fluor 750 (ThermoFisher, #S21384). Upon completion of mIF assays, sections were counterstained with Hoechst 33342 (ThermoFisher, #H3570, 1:100) and mounted with prolonged gold anti-fade medium (ThermoFisher, #P10144). 

Automated multiplex immunofluorescence was performed using a Leica Bond RX III Autostainer (Leica Biosystems, Wetzlar, Germany) to label for various clusters of differentiation (CD) markers together with NQO1, NRF1 and NRF2, visualized with TSA Cyanine 3, TSA Cyanine 5, TSA fluorescein and streptavidin-conjugated Alexa Fluor 750 respectively, similarly applied as in [10]. CD markers included CD3 (Agilent, #A0428), CD4 (Human protein atlas, #HPA004252), CD8 (Agilent, #M7103), CD20 (Agilent, #M0795), CD25 (Human protein atlas, #HPA054622), CD56 (Cell signaling technology, #3576S), CD68 (Abcam, #Ab213363, Cambridge, UK), and CD163 (Abcam, # Ab182422).

The Zeiss Axio Scan Z1 slide scanner was used to acquire fluorescence whole slide images. A uniform scanning profile was used to scan across the patients and control sections. A uniform scanning profile was created using five different fluorescent channels, including DAPI, Fluorescein, Cy3, Cy5, and Alexa Fluor 750.

### 2.3. Bioimage Analysis

Fluorescence whole slide images were analyzed using Indica HALO^®^ and HALO AI TM image analysis platforms (v. 3.4.2986.209). Firstly, HALO AI TM (https://www.indicalab.com/halo-ai) accessed on 8 September 2022, which has advanced built-in deep-learning neural network algorithms, was enabled to classify tumor regions out of tumor microenvironment areas (TME) in whole slide images by training with examples of pan-cytokeratin-positive areas under the Alexa Fluor 750 channel only. Secondly, HALO AI TM was used to develop a customized nuclear segmentation classifier, which was useful to negate falsely segmented nuclei and is often misclassified by autofluorescence in FFPE tissue sections.

Thirdly, High Plex FL (v4.1.3) module in HALO^®^ was utilized to classify the cells by the status of co-localizing NQO1, NRF1 and NRF2 by intensity thresholds of Fluorescein (cytoplasm = 5500), Cy3 (nucleus and cytoplasm = 4000), and Cy5 channel (nucleus and cytoplasm = 6000), respectively, which was similarly applied as in [34]. In total, eleven different phenotypes were classified: NQO1^+^NRF1^+^NRF2^+^, NQO1^+^NRF1^−^NRF2^−^, NQO1^+^NRF1^−^NRF2^+^, NQO1^−^NRF1^+^NRF2^+^, NQO1^−^NRF1^−^NRF2^+^, NQO1^−^NRF1^+^NRF2^−^, NQO1^+^NRF1^+^NRF2^−^, NQO1^−^NRF1^−^NRF2^−^, NQO1^+^, NRF1^+^, and NRF2^+^ and their densities were measured not only in tumor regions but also in TME (stromal and immune) regions together with the average of each dye intensity.

### 2.4. Statistical Analysis

Populations of tumor and immune cells that were single, double, or triple positive for NQO1, NRF1 or NRF2 were exported from HALO. Each cell metric, e.g., the NQO1^+^NRF1^+^NRF2^−^ tumor cell population, was then normalized to individual patient samples by dividing by the total population of tumor cells from each patient. X-Tiles biomarker assessment software was then applied to generate outcome-based cut-point optimization from the individual cell metrics, dividing populations into NQO1 high/low, NRF1 high/low and NRF2high/low for immune and stromal, and tumor cell readouts [35]. Kaplan–Meier survival analysis, Wilcoxon matched-pairs signed rank test, Spearman correlations and the Mantel–Cox test were generated using GraphPad software (Prism 9). 

## 3. Results

### 3.1. NQO1, NRF1 and NRF2 Expression in Normal Lung, Immune, Stromal and Cancer Cells

A total of 162 stage I and II adenocarcinoma NSCLC samples meeting the criteria were included in this study. Immunohistochemistry (IHC) was performed to quantify NQO1 expression in the cytoplasmic compartment of the cells after deciding the optimal dilution of the NQO1, NRF1 and NRF2 antibodies by assessing specificity, consistency, and signal-to-noise ratio [36]. Individual IHC staining of NQO1, NRF1 and NRF2 enabled the observation of protein expression by DAB chromogen brown color intensity within normal cell types, such as bronchial epithelial cells, alveoli, mucosal glands, lymphoid cells, nerve cells, endothelial cells, fibroblasts, muscle cells and arteries (Figure 1), and cancer cells. Immunostaining showed that NQO1 was positive in bronchial epithelial cells and endothelial cells, demonstrating increased NQO1 expression in ciliated surfaces when compared to the cytoplasm of columnar epithelial cells and basal cells (Figure 1A). NRF1 expression in bronchial epithelia was negative; however, NRF2 showed moderate cytoplasmic expression in columnar epithelial cells, cilia, and negative basal cells. In endothelial cells, NQO1 was strongly expressed; however, in mucous glands, lymphoid, neural, muscle and fibroblast cells, NQO1 was lowly expressed. Contrastingly, NRF2 protein was positive in lymphoid cells, and NRF1 was detected in some lymphoid cell subtypes, as shown in Figure 1. NRF1 was detected in endothelial cells; however, negative across fibroblasts, muscle, arteries and arterioles, neural cells, and mucous glands. NRF2 was present in alveolar, neural, endothelial and mucosal cells, with negative reactivity in fibroblasts. 

IHC is, at best, a semi-quantitative technique and is limiting for protein co-localization analysis. Therefore, multiplexed immunofluorescence (mIF) was performed to visualize NQO1, NRF1, NRF2, and pan-cytokeratin simultaneously. Pan-cytokeratin was utilized to classify tumors and TME regions by a deep learning neural network algorithm, DenseNet AI using Indica Labs’ HALO platform. mIF enabled the quantitative assessment of NQO1 expression within tumor and TME regions [34]. Furthermore, to distinguish several different immune cell types and to measure NQO1, NRF1 and NRF2 expression, mIF was performed again by labeling NQO1, NRF1, NRF2 together with a cluster of differentiation (CD) markers; CD3–pan T-cell receptor, CD4–helper T-cell, CD8–cytotoxic T-cell and CD25–activated T-cell, CD20- B cells, CD56- natural killer cell, CD68-pan-macrophage and CD163-dendritic cells. NQO1 protein expression was negative across all lymphoid cell subtypes (Figure 1C,D). NRF1 expression was observed in CD8^+^ and CD25^+^ T-cells, CD20^+^ B cells, CD68^+^ macrophages and CD163^+^ dendritic cells, and negative in CD4^+^ and CD56^+^ cells (Figure 1C). NRF2 expression was observed in CD20^+^, CD25^+^, CD68^+^ and CD163^+^ cells. However, NRF2 was low in CD3^+^ cells and negative in CD4^+^, CD8^+^, CD56^+^ cells (Figure 1C,D). Colocalized expression of NRF1 and NRF2 was observed in CD68^+^ cells.

Immunohistochemistry has helped to reveal that NQO1 is highly expressed in normal bronchial epithelium and endothelial cells, not in immune or stromal cells (Figure 1A,E). Multiplexed immunofluorescence, moreover, enabled us to measure NQO1 protein expression quantitatively on a single cell level of bronchial epithelial cells and cancer cells within each sample and across the cohort (Figure 1F). Notably, NQO1 protein expression was significantly higher in normal bronchial epithelial cells than in tumor cells across paired samples.

### 3.2. NQO1 Expression Is Heterogenous in Cancer

NQO1 protein expression in tumor cells labelled with pan-cytokeratin was quantitatively measured across this early-stage naïve lung adenocarcinoma cohort (I and II) and demonstrated varying levels of NQO1 on an individual cell level (Figure 2). As seen in Figure 2, pan-cytokeratin-positive tumor cells in the first row showed over 9000 signal intensity of the FITC channel, which was used to visualize NQO1, while the tumor cells in the second row had less than 1000 signal intensity. The tumor cells in the third row highlighted the heterogeneity of NQO1 abundance, demonstrating mixed FITC signal intensity values ranging between 1000 and 9000. The quantitative measurement of FITC channel intensity was divided into three categories, NQO1 high (>9222), moderate (1145≤ and ≤9222), and low (<1145). The signal intensity of the FITC channel in Figure 2 represents intra- and inter-heterogeneity of NQO1 expression in tumor cells.

### 3.3. NQO1 Protein Expression Is Independent of NRF1 and NRF2

Multiplexed immunofluorescence staining performed showed the heterogeneity of NQO1, NRF1 and NRF2 proteins within the same representative patient sample under low and high resolution, respectively (Figure 3A–C). Figure 3B shows the tissue region displaying high NQO1 expression in pan-cytokeratin-positive cancer cells, with negative NQO1 expression observed in stromal cells and immune cells. Conversely, NRF1 and NRF2 expression in these same cancer cells were lower than the expression of NRF1 and NRF2 in lymphoid cells, which are high, as seen in Figure 3B).

Figure 3C represents an NQO1 low region within the same patient sample, displaying low NQO1 expression in both tumor and stromal cells, but contrasting high NRF1 and NRF2 expression within tumor cells. 

High NQO1 protein expression in tumor cells was associated across NRF1^low^ and NRF2^low^ regions, whereas low NQO1 expression is demonstrated in tumor cells detailing NRF1^high^ and NRF2^high^ expression. 

The correlations between NQO1 and NRF1 or NRF2 were evaluated as previously shown in mouse model research, suggesting a positive correlation between NRF2 and NQO1 expression [20]. Averaged fluorescent signal intensities of NQO1, NRF1 and NRF2 from the whole cohort were plotted using Spearman’s correlation coefficient (Figure 3D,E), which demonstrated no correlative relationship between NQO1 and NRF1 (r = 0.002) or NQO1 and NRF2 (r = −0.161) expression in cancer cells.

### 3.4. Co-Localization of NRF1 and NRF2 in Cancer Cells Is Associated with Higher Patient Probability of Survival

NRF1, NRF2 and NQO1 expression in immune cells within the human NSCLC TME remain undetermined, and ROS may affect immune cell effector functions as described in MDSC mouse literature [14]. Therefore, multiplex immunofluorescence was performed to examine the co-localization of NQO1, NRF1 and NRF2 within the tumor and TME. The subsequent bioimage analysis allowed for the classification of eleven different phenotypes and density analyses of single positive cells and co-localizing cells by phenotype, e.g., NQO1^−^NRF1^+^NRF2^−^, NQO1^+^NRF1^+^NRF2^−^, and NQO1^+^NRF1^+^NRF2^+^.

The density of co-localized cell phenotypes was calculated in two separate regions-tumor and tumor microenvironment (TME) and prognostic values were assessed by overall survival metrics over five years following tumor resection. A density-based co-expression analysis was performed, which demonstrated a significant positive association between survival probability and NRF1 and NRF2 protein co-expression in cancer cells. 

Samples with high populations of NRF1^+^NRF2^+^ tumor cells displayed a significantly higher probability of survival than samples with low populations of NRF1^+^NRF2^+^ tumor cells (Figure 4, HR, 1.655; 95% CI, 1.043–2.624, * *p* = 0.03). Individual protein analysis for NQO1, NRF1 and NRF2 and triple-negative phenotyping (NQO1^−^NRF1^−^NRF2^−^) failed to generate clinically-relevant high and low cut-offs, showing no significance in tumor cells. Individual NQO1 expression in cancer cells did not show associations with the patient probability of survival. NRF1^+^NQO1^−^NRF2^−^ cancer cell populations were not indicative of improved patient survival probability (*p* = 0.15).

## 4. Discussion

NQO1 is highly expressed in breast and lung cancer tissue and has been described as a potential therapeutic target for NSCLC patients [37]. Siegel and colleagues previously described high levels of NQO1 expression in the respiratory epithelium [23]. The NRF1 and NRF2 transcription factors perform multifaceted roles in cell homeostasis and function as key regulators of ROS through the upregulation of antioxidant genes through the ARE, including nqo1, in mice and human keratinocytes [19,35,36,37]. Recent NSCLC studies demonstrate NQO1 and NRF2 negativity in normal tissue and NQO1 and NRF2 upregulation in tumor tissue, proposing single-negative NQO1 expression and dual-negative expression of NRF2 and NQO1 are predictive of improved outcomes in NSCLC patients [22,29].

In this study, NQO1, NRF1 and NRF2 were established in normal lung tissue, describing protein expression in bronchial epithelial cells, alveolar cells, mucous glands, lymphoid cells, neural cells, endothelial cells, fibroblasts, muscle cells and adjacent arterioles. This study has shown high expression of NQO1 in the respiratory epithelium in a closely-related adenocarcinoma cohort. High levels of NQO1 in cilia and decreased expression in basal cells indicate the protective role of NQO1 against environmental stressors, such as cigarette smoke, as previously described by Siegel and colleagues [23]. Similar NRF2 immunostaining of bronchial epithelium was observed at the apical surface just below cilia, affirming previously described NRF2-mediated regulation of primary ciliogenesis and potential Hedgehog signaling in mice [38].

Within the TME, excessive ROS are utilized by immunosuppressive cells, such as MDSCs and tumor-associated macrophages, against immune effector cells [14]. Highlighting insufficient research information addressing the ROS coping mechanisms of immune cells and other cells found within the NSCLC TME. Therefore, IHC and mIF were applied to investigate the expression of NQO1, NRF1 and NRF2 in immune cells and other cell types within the NSCLC TME. Immunostaining demonstrated no NQO1 in lymphoid cells, further confirmed by mIF staining of several lymphocyte subtypes. NRF1 and NRF2 immunostaining was observed in several immune cell subtypes. mIF demonstrated NRF1 and NRF2 expression in dendritic cells and macrophages, with most NRF1 expression identified in activated T-cells and the highest NRF2 expression shown in B cells. NRF1 has been reported to regulate the genetic exhaustion program during CAR T-cell therapy and increased NRF1 co-expression with NRF3 is a characteristic of exhausted T-cells (expressing PD-L1, CTLA4 and LAG3), suggesting that NRF1^+^ activated T-cells could be experiencing excessive ROS and may utilize NRF1 as potential ROS-coping mechanisms against exhaustion [39].

NRF2 has previously been shown to regulate chronic lung inflammation through the maturation of dendritic cells and by producing proinflammatory cytokines for priming B cell response to non-typeable *H. influenzae* (NTHI)-induced airway inflammation in mice [40]. Furthermore, increased NRF2 expression in alveolar macrophages and type II pneumocytes has been correlated with better incidence-of-recurrence primary spontaneous pneumothorax [41]. These findings indicate that NRF2 may be exerting a protective effect in normal respiratory epithelium and lung cancer, particularly in this early-stage cohort. NRF2 has been purported to induce IL-17D to recruit natural killer cells, mediating tumor regression in a murine melanoma model [42]. However, few CD56^+^ natural killer cells were observed in this human cohort following mIF, highlighting variance between NRF2 model systems.

In cancer cells, NQO1 expression is heterogenous, displaying low, moderate and high expression within individual cases and, importantly, between analyses, potentially due to the patient genetic background or ubiquitous ROS sources within the tumor [43]. Therefore, it highlights the difficulties of assessing NQO1 using tissue homogenization techniques and the complexities of utilizing NQO1 as a therapeutic target, owing to the differential expression of this protein within the same tumor. Examining ROS within the tumor presents further complexities, owing to the short cellular half-life associated with ROS sources and targets, such as hydrogen peroxide (H2O2, half-life ≤ 1 ms) [44].

NQO1 expression was not correlated to paired NRF1 or NRF2 expression in cancer, immune or stroma cells, contrary to studies proposing NRF1- or NRF2-directed adaptive response upon NQO1 transcription in mice and human keratinocytes. This suggests alternative transcriptional regulation of NQO1, potentially through other ARE-binding proteins such as NRF3 [15]. In cancer cells, NRF3 has also been shown to suppress NRF1 translation through the cytoplasmic polyadenylation element binding protein 3 (CPEB3) [30]. NRF1 repression redirects ubiquitin-dependent protein degradation from the NRF1-26S proteasome regulation axis towards the proteasome maturation protein (POMP)-20S proteasomal axis. Thereby facilitating retinoblastoma and p53 protein degradation and subsequent cancer growth [30]. The intrafamilial regulation between NRF1, NRF2 and NRF3 remains to be elucidated, and many NRF3 transcriptional targets remain uncharacterized. Due to a lack of robust commercially available NRF3 antibodies, assessing NRF3 protein expression within this cohort was out of the scope of this paper, thereby limiting the characterization of ROS-coping mechanisms of immune cells. Furthermore, the significantly shorter respective half-life of NRF1 and NRF2 (0.5 h and 0.25–0.5 h, respectively) when contrasted to the half-life of NQO1 (18 h) highlights the limitations of protein expression association in two-dimensional tissue sections and cells [45,46,47].

Individual NRF1, NRF2 and NQO1 expression within all cancer, immune and stromal cells are heterogenous in this closely-related cohort. However, increased double-positive NRF1 and NRF2 expression in cancer cells are associated with an improved probability of survival over five years. This association may align with mouse model findings describing the anti-inflammatory effects of NRF2 activation, which antagonizes tumor-promoting inflammation and suppresses cancer malignancy [24,48]. To establish the potential prognostic value of NRF1 and NRF2 in the NSCLC TME, the expression of NRF3 requires quantification and further functional assessment. Additional investigation into ROS-coping mechanisms would require population and spatial-based analysis, paired with an antioxidant protein mIF panel targeting immune and immunosuppressive cell subtypes within the TME.

This study demonstrates that NQO1 protein expression is high in normal, tumor-adjacent tissue and that NQO1 expression varies depending on the cell type. Inter and intra-patient heterogenous NQO1 expression was observed in lung cancer, displaying NQO1 expression are independent of NRF1 and NRF2 in tumors. NRF1 and NRF2 double-positive expression in cancer cells was associated with the improved patient probability of survival.

## Figures and Tables

**Figure 1 biomolecules-12-01652-f001:**
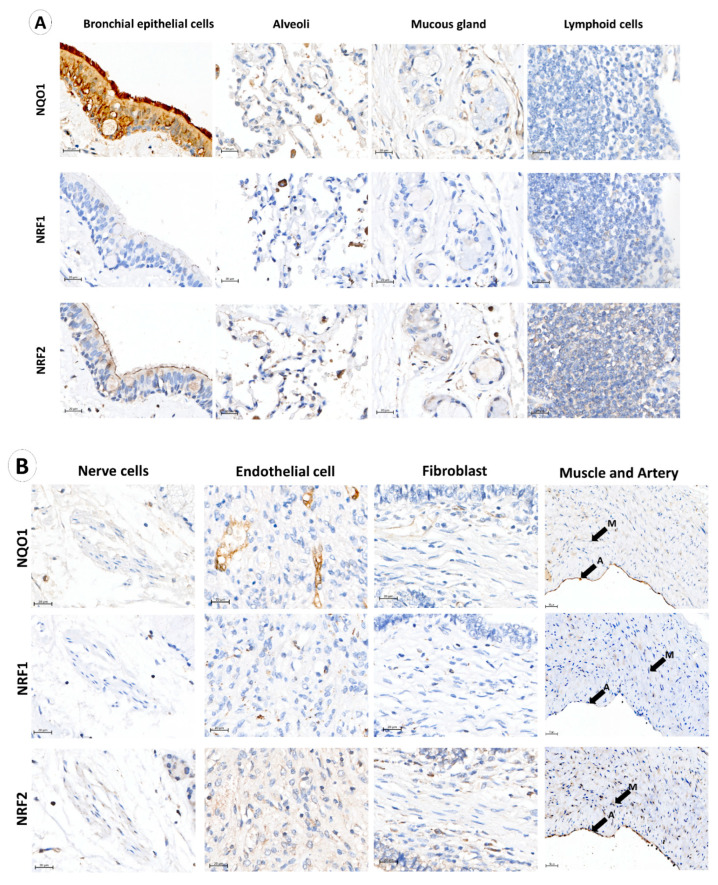
NQO1 protein abundance in normal lung tissue. (**A**) NQO1, NRF1 and NRF2 immunohistochemical staining demonstrated normal bronchial epithelium, alveolar epithelium, mucosal gland, and lymphoid cells. (**B**) NQO1, NRF1 and NRF2 expression in nerve cells, endothelial cells, fibroblast cells and muscle (M) with the adjacent artery (**A**). (**C**,**D**) Multiplex immunofluorescence of lymphoid cell types adjacent to tumor tissue using DAPI (blue nuclear stain), cluster of differentiation markers (yellow), NRF1 (green), NQO1 (red) and NRF2 (purple). Composite images display DAPI, CD markers, NRF1, NQO1 and NRF2 overlay. (**E**) NQO1 (green stain) and pan-cytokeratin (pink cell structure stain highlighting bronchial epithelium and cancer cells). Composite images display DAPI, NQO1 and pan-cytokeratin overlay. N denotes normal bronchial epithelial cells, and C denotes cancer cells. (**F**) NQO1 intensity-based analysis displaying paired average NQO1 cell intensity as a surrogate for NQO1 protein expression in normal bronchial epithelial cells and tumor cells within each patient sample. Data points designate patient samples out with 10th and 90th percentiles. Statistical significance was assessed using the Wilcoxon matched-pairs signed rank test (*p* **** < 0.001).

**Figure 2 biomolecules-12-01652-f002:**
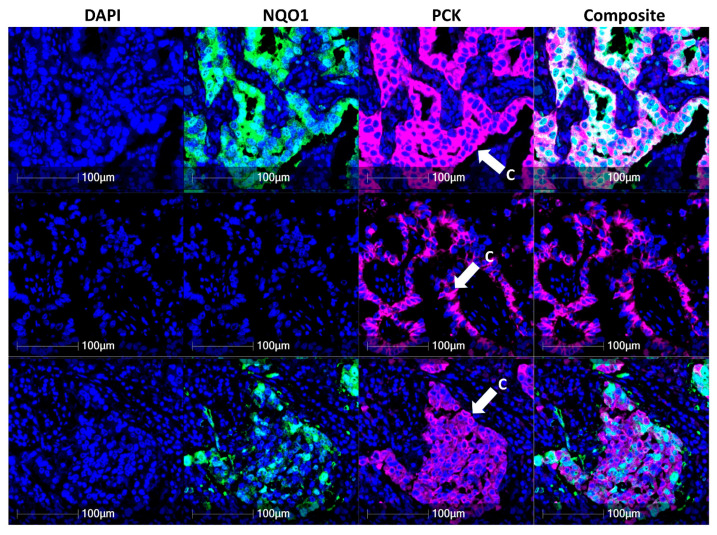
NQO1 heterogeneity in lung adenocarcinoma. The expression of NQO1 in cancer cells was examined across the NSCLC cohort using DAPI (blue nuclear stain), NQO1 (green) and pan-cytokeratin (pink cell structure stain highlighting bronchial epithelium and cancer cells). Composite images display DAPI, NQO1 and pan-cytokeratin overlay. C denotes cancer cells.

**Figure 3 biomolecules-12-01652-f003:**
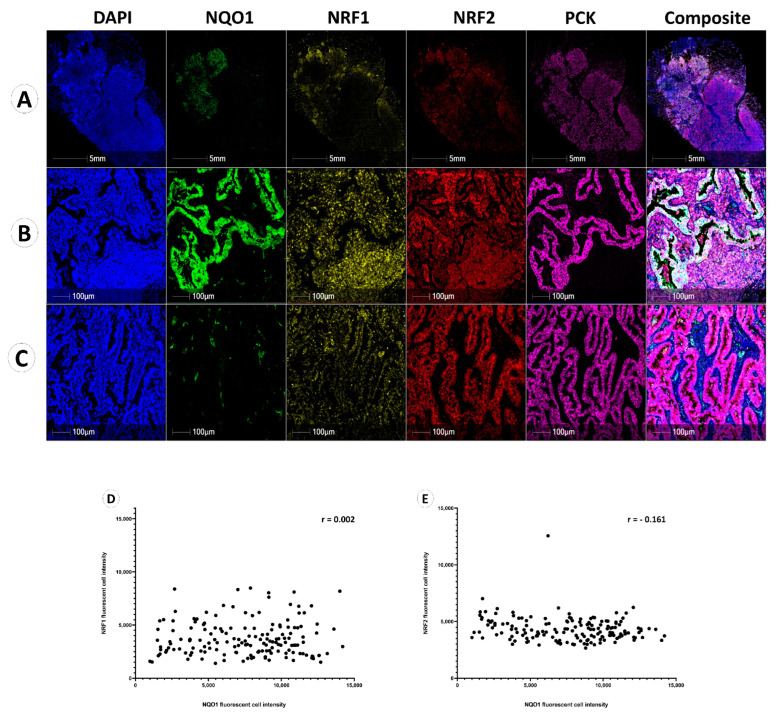
NQO1, NRF1 and NRF2 expression in NSCLC. Five fluorescent channels illustrating the expression of DAPI (blue), NQO1 (green), NRF1 (yellow), NRF2 (red), and pan-cytokeratin (pink), individually. (**A**) Whole slide image (WSI) of a case showing heterogeneous NQO1 expression. Composite images display all channels’ overlay. (**B**) NQO1 highly expressed region in WSI (**A**). (**C**) NQO1 lowly expressed region in WSI (**A**). (**D**) Spearman correlation between NRF1 fluorescent cell intensity and NQO1 fluorescent cell intensity in cancer cells (r = 0.002). (**E**) Spearman correlation between NRF2 fluorescent cell intensity and NQO1 fluorescent cell intensity in cancer cells (r = −0.161).

**Figure 4 biomolecules-12-01652-f004:**
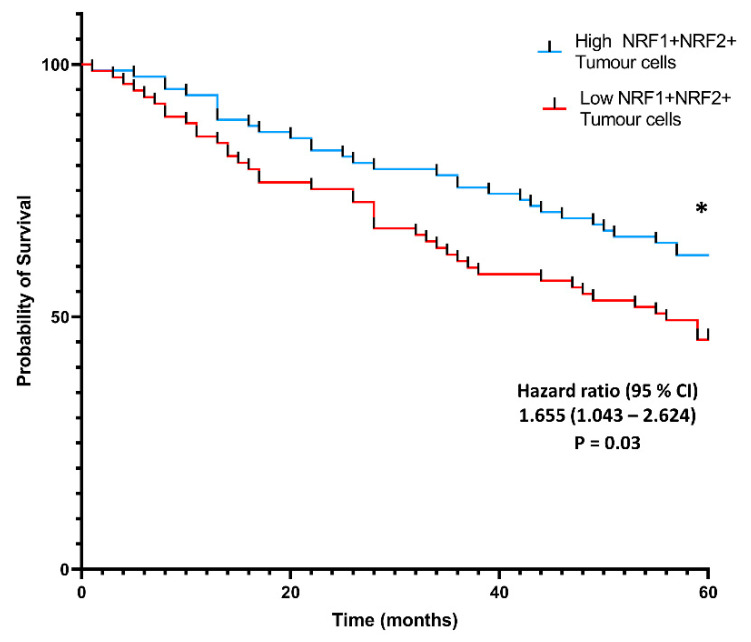
Kaplan–Meier survival analysis of the density of NRF1 and NRF2 co-expressed cancer cells. Population-based Kaplan–Meier five-year survival analysis demonstrates the association between NRF1^+^NRF2^+^ co-expressing tumor cells and the probability of survival. Statistical significance was assessed using the log-rank test (Mantel–Cox). (*p* * < 0.05).

## Data Availability

The data presented in this study are available on request from the corresponding author.

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
