# Peer review of "Spatial Analysis of NQO1 in Non-Small Cell Lung Cancer Shows Its Expression Is Independent of NRF1 and NRF2 in the Tumor Microenvironment"

_biomolecules, 2022, doi:10.3390/biom12111652_

Round 1
Reviewer 1 Report
The manuscript "NQO1, NRF1 and NRF2 in the tumor microenvironment of human non-small cell lung cancer" describes the combined expression of NRF1 and NRF2 to be associated with longer survival of non-small cell lung cancers. Authors also indicate that the expression of NQO1 is heterogenous and is independent of NRF1 and NRF2 expression. Overall, authors have presented evidence for their conclusions. The Title needs a change to reflect their conclusions.
Minor corrections
Fig. 1B, IHC in Muscle and arteriole: Expression of NQO1 in arterioles need better representation. Higher magnification of the signal area within this figure will be helpful.
Figure 3D and E. The spearman correlation in the figures and figure legend indicates NRF1 vs NQO1 to be r=0.002, and NRF2 vs NQO1 to be r=-0.161. However, description of this result in lines 322 to 324 is reversed.
Line 157: Antibody NR2 (Protein tech # 16396-1-AP) should be NRF2 (Protein tech # 16396-1-AP)
Reference 39 is incomplete: Journal title, volume, page numbers and year (Gennert DG et al. Dynamic chromatin regulatory landscape of human CAR T cell exhaustion. Proc Natl Acad Sci U S A. 2021 Jul 27;118(30): e2104758118. doi: 10.1073/pnas.2104758118.Proc Natl Acad Sci U S A. 2021. PMID: 34285077 Free PMC article) are missing.
Author Response
Dear Reviewer,
We have made the necessary amendments to our manuscript as per Author corrections and comments:
- The manuscript title has been changed.
- Fig 1. B) Figure has been changed to demonstrate arterial endothelial cells and muscle more clearly.
- 3 D and E) r values in the text have been corrected to correspond as described in the respective figure.
- Line 157: Full protein name has been amended.
- Reference 39 has been fully populated.
Thank you.
Reviewer 2 Report
I have read with interest your manuscript, “NQO1, NRF1 and NRF2 in the tumor microenvironment of human non-small cell lung cancer”, reporting data on the significance and biological role of NQO1 and the two transcription factors with antioxidant activity, NRF1 and NRF2. I consider your data interesting and worthy of being accepted for publication in a scientific journal of Biomolecules level.
Author Response
Thank you for your review.
Reviewer 3 Report
This paper talks about that NQO1 protein expression is high in normal, tumor-adjacent tissue and that NQO1 expression varies depending upon the cell type in ques-tion. Inter and intra-patient heterogenous NQO1 expression was observed in lung cancer, displaying NQO1 expression is independent of NRF1 and NRF2 in tumors. NRF1 and NRF2 double-positive expression in cancer cells was associated with improved patient probability of survival. Overall, the paper is suitable for publication in present form.
Author Response
Thank you for your review.